# Management of Casualties from Radiation Events

**Robert Alan Dent** 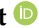

85th WMD Civil Support Team, Utah Army National Guard, 1640 North 2200 West,
Salt Lake City, UT 84116, USA; robert.a.dent.mil@army.mil; Tel.:+1-801-520-1330

**Abstract:** Radiation events such as nuclear war, nuclear reactor incidents, and the deployment of a radioactive dispersal device (dirty bomb) are all significant threats in today's world. Each of these events would bring significant challenges to clinicians caring for patients with burns and traumatic injuries who are also contaminated or irradiated. The result of a nuclear exchange in a densely populated area could result in thousands of patients presenting with trauma, burns, and combined injury (trauma and burn in an irradiated patient). In this review, we will discuss the three major types of ionizing radiation: alpha, beta, and gamma, and their respective health hazards and biological effects. Additionally, we will discuss the types of burn injuries in a nuclear disaster, caring for the contaminated patient, and managing the combined injury of burn trauma with acute radiation syndrome. The reader will also be left with an understanding of how to prioritize lifesaving interventions, estimate the absorbed dose of radiation, and predict the onset of acute radiation syndrome. While some animal models for morbidity and mortality exist, there is limited modern day human data for patients with combined injury and burns associated with a nuclear disaster due to the infrequent nature of these events. It is extremely important to continue multidisciplinary research on the prevention of, preparedness for, and the response to nuclear events. Furthermore, continued exploration of novel treatments for radiation induced burns and the management of combined injury is necessary.

**Keywords:** radiation; acute radiation syndrome; radiation burn; decontamination; nuclear war; nuclear reactor; radiological dispersal device

## 1. Introduction

Nuclear weapons, nuclear reactor incidents, and radioactive dispersal devices (RDDs) all pose a significant threat and subsequent challenges for healthcare professionals and facilities in today's world. The deployment of the world's first nuclear weapons in Japan in 1945 and the Chernobyl accident of 1986 are some of the few case studies we have. Clinical information from these events is complicated due to incomplete data gathering at the time and profound differences in medical technologies and capabilities between then and now. There are many smaller scale incidents involving the medical effects of ionizing radiation which are not related to war but are excellent case studies and worth reading by clinicians in order to better understand radiation injury. This article will discuss the three most likely radiation threats subsequent to conflict: the deployment of a nuclear weapon, a nuclear reactor incident (criticality), and a RDD or "dirty bomb". It is important to know the differences between these threats as they pose different hazards and medical implications. Paramount to understanding radiation injury is a knowledge of basic radiation physics, as the type of radiation and dose absorbed can dictate different approaches to medical care.

## 2. Radiation Physics

Multiple forms of radiation are present during a nuclear event. Electromagnetic energy forms include gamma radiation and X-rays. X-rays are very similar to gamma radiation and differ only in their origin within the atom [1]. Gamma radiation is emitted from the nucleus,

while X-rays are emitted from outside the nucleus [2]. Given this small difference, the only electromagnetic radiation that is typically discussed in relation to medical care is gamma radiation. Due to its electromagnetic nature, gamma radiation does not pose a risk of contamination, unless a gamma emitting isotope is on, or embedded in a victim. There are many therapeutic uses for gamma radiation in medicine such as radiotherapy for tumors or total-body irradiation prior to hematopoietic stem-cell transplantation. Particulate radiation includes alpha particles, beta particles, and neutrons. Neutrons are produced almost entirely from fission events and released in the first few seconds of a nuclear blast, with no significant number of neutrons produced after that [2]. Neutron radiation produces the same biological effects as gamma radiation and there is sufficient evidence in animal models that it is more carcinogenic [3]. Because neutrons are present only transiently, have a biological effect similar to gamma radiation, and do not pose a contamination problem, they are rarely referred to in the context of medical treatment. The alpha particle is a positively charged and comparatively heavy decay product. Alpha particles are highly energetic but are so heavy that they are unable to travel very far from the atom. This characteristic makes them almost harmless outside of the body [2]. The major hazard of an alpha particle is when it is ingested, inhaled, or contaminates wounds which would then allow the ionization of sensitive tissue such as the gastrointestinal or respiratory tracts and wounds. Beta particles are very small, fast moving particles with a negative charge that are emitted from the atom's nucleus. Beta particles travel farther in air and penetrate more efficiently than alpha particles but can be shielded by a thin layer of clothing or simple shielding such as aluminum [2]. Although beta particles can cause radiation burns, their primary hazard is ingestion or inhalation. Alpha and beta particles will take on the form of whatever particulate they are incorporated into. This could be dust, smoke, or suspended blast debris. Alpha and beta radiation therefore become contaminants on clothing, skin, equipment, vehicles, and agricultural crops. Long term exposure occurs through the deposition of particles in the environment and subsequent ingestion of contaminated food and water. Internalized radioactive isotopes can have a devastating effect on tissue and each isotope has a unique target organ, or organs. Patients presenting to a hospital from a radiation incident site are expected to be contaminated, therefore treatment teams must be able to detect the presence of contamination using handheld radiation detectors such as a Geiger–Muller detector and then decontaminate them to levels that are as low as is reasonably achievable. Medical countermeasures, or antidotes, exist for some isotopes and will be discussed later. The process of using an antidote to decrease the isotope burden in the body is called decorporation.

Although not related to a nuclear event, there are interesting case studies of alpha and beta radiation contamination causing significant clinical effects and death in recent history. In 2006, Alexander Litvinenko was intentionally poisoned in London, UK, with the alpha emitter Po-210 which was deposited in a cup of tea. Mr. Litvinenko presented to the hospital with nausea, vomiting, and diarrhea. He was admitted to the hospital with the diagnosis of gastroenteritis and dehydration. Within 6 days he developed pancytopenia and by hospital week three he developed multi-organ failure and died on day 23 of his illness [4]. One of the most profound clinical cases involving a beta emitting isotope happened at the Chernobyl nuclear plant in 1986. The explosion of the reactor released large amounts of steam and water contaminated with the beta and gamma emitting isotopes Cs-134, Cs-137, and Sr-90. Employees and rescue workers became saturated in this contaminated water and 26 patients (of 237 suspected or documented to have a radiation injury) died in the first 3 months after exposure due to skin lesions of greater than 50% of the total body surface area [5]. This incident will be discussed in further detail later.

Radioactivity, absorbed doses, and exposure are referred to in either the International System of Units (SI) or in common units (Table 1). In medicine, the unit system used is often dictated by geographical traditions. The most commonly referenced unit for absorbed dose in medicine is the gray (Gy). The sievert (Sv) and rem are dose equivalent units which are also used to predict the potential health effects of ionizing radiation [6]. The acute lethal

whole body absorbed dose of radiation in humans which would be expected to cause death in 30 days without treatment (LD 50/30) is 4.5 Gy [2].

**Table 1.** Health and Human Services. Radiation Event Medical Management. Source: www.remm. hhs.gov, accessed on 12 July 2023 [7].

| International System of Units (SI) and Common Unit Terminology | | |
|---|---|---|
| | **SI Units \*** | **Common Units** |
| **Radioactivity** | becquerel (Bq) | curie (Ci) |
| **Absorbed Dose** | gray (Gy) | Rad |
| **Dose Equivalent** | sievert (Sv) | Rem |
| **Exposure** | coulomb/kilogram (C/kg) | roentgen (R) |

\* SI Units: International System of Units.

## 3. Nuclear Weapons

Nuclear weapons detonate by nuclear fission, or a combination of fission and fusion using weapons-grade plutonium or uranium. The energy released from a nuclear detonation is exponentially higher than even the largest conventional explosives, and modern-day weapons may approach yields of up to 500 kilotons of TNT. As a frame of reference, the Hiroshima nuclear detonation in August 1945 was equivalent to approximately 15–20 kilotons of TNT [6]. Response plans are commonly modeled on the deployment of a 10-kiloton weapon, as this represents the most likely scenario from a ground detonation of a tactical nuclear or terrorist weapon. The approximated distances and zones mentioned below are based off of a 10-kiloton detonation. If a much larger weapon is deployed the distances and zones would be much larger, and the yield of a weapon is the most important factor in determining the level of casualties and damage [2].

The sequence of events following nuclear detonation generates casualties through multiple processes. The detonation will produce an immediate large, hot fireball, thermal radiation, prompt nuclear radiation, air blast wave, residual nuclear radiation, electromagnetic pulse (EMP), interference with communication signals, and, if the fireball interacts with the terrain, ground shock [2]. In the severe damage zone (radius of 130 m) there will be immeasurable heat, similar to temperatures at the center of the sun, and a high blast overpressure (7–9 psi) causing extreme infrastructure damage [2]. Due to the nuclear fission process, large amounts of gamma radiation and neutrons are produced in the first minute; this is referred to as prompt nuclear radiation. Prompt radiation doses are estimated to be anywhere from 125 cGy to 3000 cGy [2,8]. Victims in this vicinity will have almost no chance of survival. A thermal pulse expands outward at the speed of light, while the shock wave can travel at a few seconds per mile. A moderate damage zone will be 0.5 to 1 mile from ground zero. Blast overpressures are 2–5 psi and the infrastructure will be structurally unstable but standing, with a predominance of structural fires [8]. This zone will generate significant casualties from blast, blunt, and penetrating trauma; thermal burns; and radiation burns and exposure. Thermal burns are the primary cause of burn injury because the thermal envelope extends well beyond the radiation contours [9]. The rising fireball will pull thousands of tons of dirt and debris into the atmosphere. Radioactive alpha and beta particles will become incorporated into this debris, becoming a plume of contamination which can be thousands of feet high and be dispersed with the prevailing wind pattern. In the hours to days following detonation, structural fires will spread outward several miles from ground zero, and hazardous levels of radioactive fallout will be deposited downwind. Depending upon weather patterns and precipitation, low levels of radioactive fallout can be deposited globally over a period of weeks [10].

## 4. Types of Burn Injury in a Nuclear Disaster

Burn injury will be a significant challenge for clinicians following a nuclear detonation and will be complicated by patients presenting with combined injury, i.e., irradiated patients also having mechanical trauma and burns. It Is estimated that burns caused 50 percent of the deaths at Hiroshima and Nagasaki [11]. A unique feature of this weapon is that an appreciable fraction of the liberated energy goes into radiant heat and light. This very short duration of heat radiation, just a few thousandths of a second, leads to the development of flash burns [11]. Flash burns are thermal burns and will be limited to exposed skin facing the blast and will vary in severity depending upon distance from the blast and duration of exposure [12]. Figure 1 shows examples of flash burns from Japan in 1945.

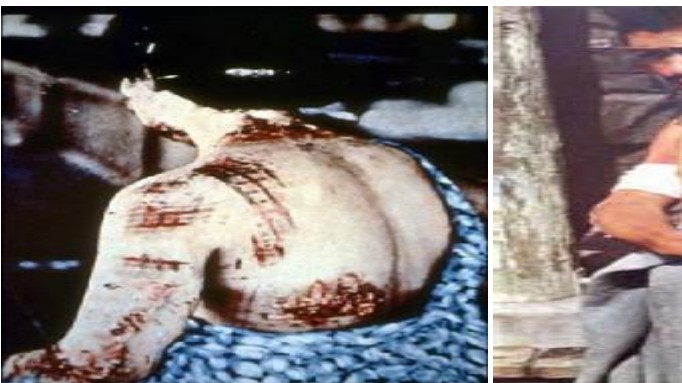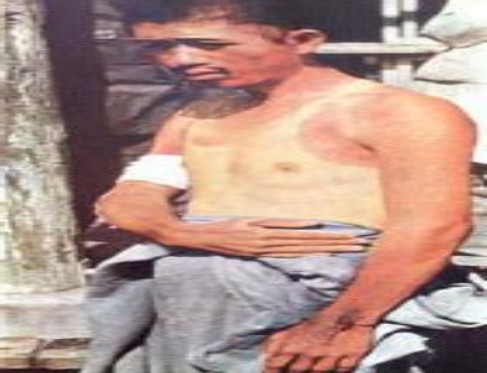

**Figure 1.** The figure on the left demonstrates thermal energy absorption by darker areas of clothing, the figure on the right shows a profile burn in which the skin was spared under light colored clothing which reflected the thermal energy [8]. Pictures courtesy of Planning Guidance for Response to a Nuclear Detonation, 3rd Ed. Federal Emergency Management Agency. May 2022, p. 75.

It is estimated that a 1 megaton explosion could cause full thickness flash burns at distances of up to 8 km, partial thickness burns up to 10 km and superficial burns up to 11 km from ground zero [11]. In addition to flash burns, thermal injuries caused by secondary fires are expected. It is important to note that any burn seen acutely after the incident is almost always a thermal burn, to include flash burns. Differentiation between flash and traditional thermal burns may be made by the patient not having a history of other thermal exposure, or the signature patterns associated with thermal energy absorption by clothing as depicted in Figure 1. Cutaneous injuries that are caused by beta or gamma radiation will almost always be delayed in onset.

Both beta contamination and whole or partial body gamma radiation and neutron exposure can cause cutaneous radiation syndrome (CRS). When the radiation injury to the skin is more localized or the radiation dose is insufficient to penetrate to deeper organs it may be referred to as cutaneous radiation injury (CRI) [13]. While CRS is a subset of acute radiation syndrome, it can occur in isolation of the of other 3 subsets (gastrointestinal, hematopoietic, neurovascular). In the case of a beta contamination burn, it is less likely that the energy transmission into the tissue would penetrate deep enough to cause the other three subsyndromes of ARS [13]. CRS can be a spectrum of presentations ranging from transient erythema and pruritis to a full-thickness burn. As previously stated, the important difference between CRS and thermal burns is that while thermal burns are apparent acutely, CRS may take days to weeks to manifest, depending upon the absorbed dose, quality of the radiation, and the characteristics of the target cells responsible for a given lesion. It is estimated that an acute absorbed dose of 350–500 cGy would be required to produce CRS. CRS is also associated with a small but real potential for malignancy as a late effect, and chronic scarring patterns are different [14]. CRS should be managed in a dedicated burn unit, if possible.

CRS will occur in 4 phases, also seen with other target organs of ARS: prodromal, latent, manifest illness, and recovery. The prodromal phase can include itching, tingling, and

transient erythema or edema. Treatment is aimed at symptom control with antipruritics, antihistamines, and topical anti-inflammatory therapy [13]. Care should be taken to carefully document skin lesions using sketches or photographs. A latent, symptom-free phase may last a few days to a few weeks. Depending upon the suspected extent of a lesion, other treatment considerations include antimicrobial prophylaxis and treatment, inhibitors of proteolysis, growth factors to enhance granulation and re-epithelialization, and stimulation of the local blood supply with pentoxifylline [15]. Multiple novel countermeasures for CRS and CRI are being evaluated in animal models, these include aCT1 peptide, thrombin peptide, mesenchymal stromal cells, and angiotensin analogues [16]. Following the latent phase, a manifest illness phase will occur due to an irradiated basal layer of the skin. Signs and symptoms will include inflammation, erythema, dry or moist desquamation, ulceration, blistering and epilation. Severe wounds may require local excision, grafting for closure or amputation [13]. If the development of ARS is suspected, surgical interventions are most effective and successful if done early in the course of the lesion. The rationale for this is that within days to weeks the patient could develop neutropenia, thrombocytopenia, and fluid and electrolyte issues which could complicate the procedure and wound healing afterwards. Full-thickness graft and microsurgery techniques are the most effective [15].

## 5. Contaminated Wounds and Burns

Patients arriving at a treatment facility following a nuclear detonation, reactor criticality, or RDD are expected to have contamination in addition to a broad spectrum of traumatic injuries that require immediate attention. Decontamination of radiation is conducted similarly to chemical decontamination; the main difference is timing. Chemical decontamination is often an emergency, radiological decontamination is not an emergency [17]. Decontamination of the patient should never delay lifesaving medical care. Medical teams should acquire and train with personal protective equipment (PPE) that allows them to operate in a contaminated environment. This PPE should include at a minimum: a head covering, eye shield, fitted full face piece air purifying respirator (P100 or better), full-length disposable overgarment, latex or nitrile gloves, and disposable shoe covers [18]. Providers should utilize their PPE while performing triage and providing lifesaving interventions at a casualty collection point outside the receiving facility. Of note, this PPE ensemble is subject to downgrade or upgrade by incident commanders or emergency management specialists based on the operating environment. For example, if a hospital staff is caring for a patient who is minimally contaminated, respiratory protection with an N95 or simple surgical facemask may be sufficient (in addition to the above mentioned contact precautions).

Once a patient is considered stable, they can move through a decontamination corridor. The decontamination corridor should be run by personnel who are familiar with basic radiation detection equipment capable of detecting all three forms of radiation, such as a Geiger–Mueller instrument with a pancake probe. Normal background radiation readings should be obtained before patients arrive as a frame of reference. Patients are considered contaminated when they are greater than 2 times background radiation levels, or 100 counts per minute above background. It is important to note that contamination levels even several times above background pose very low risk to treatment teams in PPE; just because a patient is contaminated does not mean they pose a hazard to providers. Once a patient is deemed to be contaminated, decontamination should start by first protecting the airway and mouth with a mask to prevent the ingestion and inhalation of particles during the decontamination process. The patient's clothing, shoes, and personal effects should then be gently removed to prevent making particles airborne. This step alone will remove approximately 90 percent of contaminants. Clothing should be bagged and tagged as contaminated. The patient can then be wiped with a dry microfiber type cloth, with special attention given to the hair and exposed skin. A lightly moistened cloth or wipe can be used in areas like the face and hair, where the presence of moisture may help tack particles to the cloth. Wounds and burns should be carefully surveyed with a radiation monitor to rule out the presence of

contamination or embedded radioactive fragments. Closing wounds with gross radioactive contamination will significantly complicate wound recovery. Contaminated wounds should be irrigated with tepid water, and the waste irrigation solution should be collected and marked as contaminated [19]. Additionally, radioactive foreign bodies should be placed in a container marked radioactive and placed in an isolated area, preferably with shielding in the form of a radiology apron or other dense material. It is important to protect the patient from adverse effects such as hypothermia while they are being decontaminated [20]. Once decontamination is complete, the patient is surveyed again and considered decontaminated when readings are less than twice background. This can usually be accomplished by performing two decontamination cycles [21]. Focal areas of residual contamination can be addressed by repeated wiping with a new microfiber cloth or taking a piece of tape or other adhesive and gently pressing to the skin, wound or burn to capture radioactive particles. It is important to note that sometimes achieving less than twice background will not be possible; in general, this does not pose a risk to providers in the hospital as long as PPE is worn. The goal then becomes to decontaminate to a level that is as low as is reasonably achievable. Additionally, in a setting of heavy contamination and mass casualties, it is acceptable to tolerate a higher level of residual contamination to expedite the process. Casualties with residual contamination can be wrapped in a sheet or a blanket in a cocoon fashion throughout their care process as possible. This will shield treatment teams from any radiation present and minimize contamination in the treatment area.

## 6. Radiation Injury Impact on Triage

An absorbed dose of radiation may significantly impact mortality both in the setting of combined injury and radiation injury alone. This is because the management of burns and trauma becomes increasingly difficult in the setting of neutropenia, thrombocytopenia, and fluid and electrolyte losses secondary to ARS. We will discuss the impact of radiation on triage in the setting of both radiation injury alone, and with a combined injury such as trauma or burns. Triage can be significantly complicated by resource availability. Using the DIME model of triage (Minimal, Delayed, Immediate and Expectant), patients with radiation injury alone at a dose of less than 2 Gy are considered Minimal. Doses of 2 to 6 Gy are Immediate. Doses of 6 to 10 Gy are Immediate except in the setting of constrained resources in which these patients could be triaged as Expectant. Patients who receive 10 Gy or greater are largely Expectant [22].

Figure 2 compares triage categories of patients who have a solitary radiation injury of greater than 2 Gy with combined injury patients receiving the same dose. The 2 Gy dose is the accepted threshold to elevate the triage category by 1 level in a combined injury [23]. If a burn exceeds 20% TBSA, the triage category would be elevated by 1–2 levels. Methods of estimating absorbed radiation doses both by symptomatology and laboratory data are discussed later.

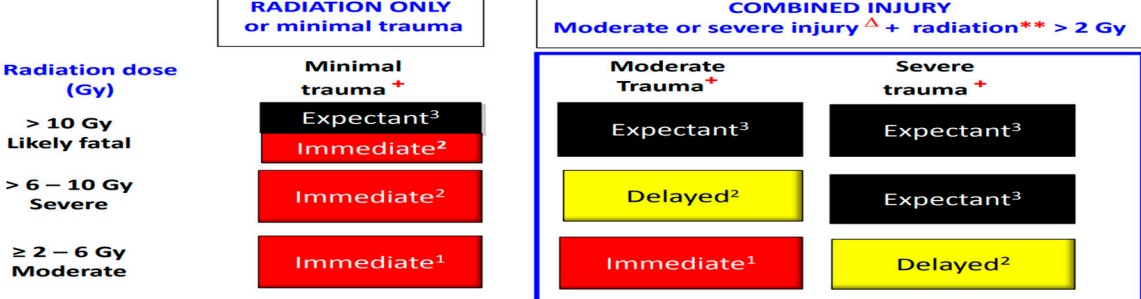

**Figure 2.** Δ indicates burns greater than 20% TBSA. ** radiation dose received by the whole body or a significant portion of the body. + trauma category. Health and Human Services. Radiation Emergency Medical Management. Source: www.remm.hhs.gov, accessed on 12 July 2023.

In the event of the deployment of a nuclear weapon, local medical resources will clearly be overwhelmed, and triage may need to be adjusted to a crisis standard of care. Burkle and colleagues described the concept of a national or international burn specific emergency medical team (EMT) which would be designed to assist with these patients to include excision and grafting in a field hospital setting [24]. In order to be effective, these teams would need to have the personnel, training, and resources in order to make them rapidly deployable.

## 7. Nuclear Reactor Incidents

In the event of a nuclear reactor incident, medical professionals will be challenged with the same types of conventional and radiation injuries as seen in a nuclear detonation, albeit on a smaller scale. Nuclear reactors are vulnerable to compromise by natural or manmade disaster, malfunction, or cyber-attack. The Chernobyl accident in 1986 is a valuable case study on the medical implications of a reactor incident. The accident happened when uranium fuel in the reactor overheated and melted through the protective barriers. This led to an explosion and fire that demolished the reactor building and released large amounts of radioactive isotopes into the atmosphere. The initial explosion resulted in the immediate deaths of 2 workers who were in the vicinity of the explosion. Twenty-seven firemen and mitigation workers died over the following 3 months of acute radiation syndrome (ARS). Victims who had ARS and burns greater than 50% TBSA had an extremely high mortality. Over 150,000 square kilometers were considered contaminated, and the community suffered a markedly higher incidence of childhood thyroid cancer in those who were 0–14 years old at the time of the accident [25]. This cluster of thyroid malignancy is attributable to the ingestion of food and milk contaminated by radioactive iodine fallout.

## 8. Radioactive Dispersal Device (RDD)

The RDD or "dirty bomb" as it is commonly referred to is distinctly different than a nuclear weapon and would be deployed as an act of terrorism. The RDD is a conventional explosive with a radioactive isotope incorporated into the device. These radionuclides are potentially available from military, medical, academic, research, and industrial sources and may be found, stolen, or purchased legally [26]. The use of any radioactive isotope is possible, however, the isotopes that are the most common in industry which could be used are Co-60, Sr-90, Cs-137, Ir-192, Pu-238, and Am-241 [27]. The world has yet to see the deployment of an RDD, but it remains a significant threat. Detonations would likely be of far lesser yield than a nuclear detonation, and do not use fission or fusion in the detonation process. The primary threat of the RDD is the conventional blast and associated traumatic injuries. Radioactive materials can be incorporated into the device in the form of a salt (such as $Cs^{137}$) or in a metal form. They are dispersed up to a few hundred meters from ground zero, and medical teams should prepare to survey patients with penetrating injury for embedded fragments. Unlike a nuclear explosion, the prompt dose of radiation released by an RDD is significantly smaller and highly unlikely to cause ARS [28]. The caveat to this is in the case of a bomb deployed in a confined space, such as a subway, in which the radiation dose could be sufficient to cause ARS to victims in the vicinity of the blast [28]. Lastly, there will be downwind fallout of very small particles which pose a contamination problem but in general would not result in adverse health effects. Because of the fear of radiation among the public, the RDD is a potent psychological weapon and is likely to cause mass panic. Communication through public health channels is paramount in clarifying the actual health risks of the radiation hazards present.

## 9. Acute Radiation Syndrome

ARS results from target organ dysfunction following whole-body irradiation. Organs which are most sensitive to radiation are those with rapidly dividing cell populations, i.e., the gastrointestinal and hematopoietic systems. The central nervous system can also be affected by very high doses of radiation in excess of 10 Gy.

In the acute phase, the gastrointestinal system is the first indicator of radiation exposure and is seen with doses of 2–4 Gy or greater (Figure 3). Nausea and vomiting are the first indicators of significant radiation exposure and use of the "time to vomiting" reference is a rudimentary way of estimating the effective dose of radiation absorbed [29]. The greater the radiation dose, the earlier vomiting will present.

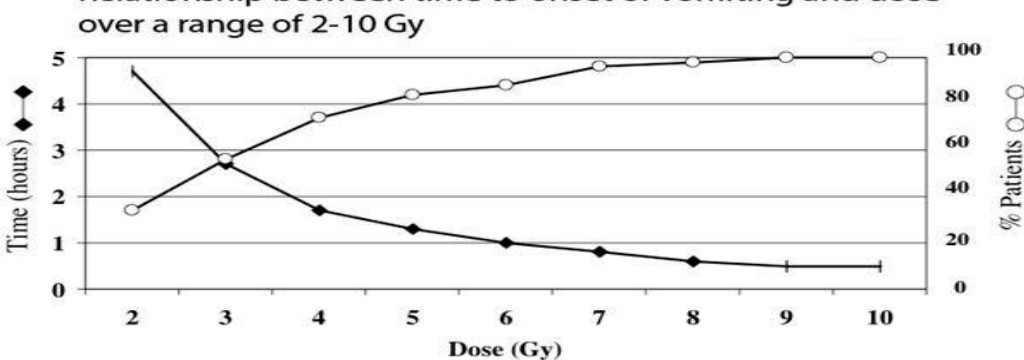

**Figure 3.** Health and Human Services. Radiation Event Medical Management. Source: www.remm.hhs.gov, accessed on 2 August 2023 [29].

Patients who develop unexplained vomiting within 1 h of the event have received at least 6 Gy and will likely have a poorer prognosis. 5HT3 receptor blockers such as ondansetron, and benzodiazepines are the first line treatment for vomiting. In refractory cases, a neurokinin-1 receptor antagonist such as aprepitant may be beneficial. Patients can also develop anorexia, abdominal pain, diarrhea, and hematochezia. Supportive care to include fluid and electrolyte replacement and loperamide is the mainstay of therapy. Radiation enteropathy is characterized by inflammation or cell death including mucosal cell loss, acute inflammation in the lamina propria, eosinophilic crypt abscess formation, and swelling of the endothelial lining of arterioles [30]. This makes it extremely difficult to digest or absorb fluid and nutrients; therefore, intravenous fluid and electrolyte replacement is favored over oral replacement. Additionally, patients with active radiation enteropathy are not likely to tolerate enteral feeding and may need total parental nutrition. There is some evidence to suggest that statins increase the level of endothelial thrombomodulin and may reduce the effects of radiation in the GI tract [31]. The manifest illness stage may last for weeks before the patient improves.

The hematopoietic syndrome is the second major complication of ARS and requires a dose of 2 Gy or greater [32]. The blood-forming tissue in the bone marrow is extremely sensitive to radiation and subsequently patients are expected to develop leukopenia, anemia, and thrombocytopenia. Of the white blood cell lines, lymphocytes are the most sensitive to radiation exposure. A second method of estimating the effective radiation dose (in addition to time to vomiting) is the calculation of lymphocyte depletion kinetics. A complete blood count with differential should be obtained at the earliest opportunity of the patient's care, and then repeated every 8–12 h. The dose absorbed is correlated with the rate of decline and nadir of peripheral blood lymphocytes as shown in Figure 4. This tool is only valuable between the time of exposure and 11 days post exposure [33].

A final method of estimating the absorbed dose is drawing a single blood sample for dicentric chromosome analysis between 24 h and 4–6 weeks after exposure. This test is mostly useful in victims with a potential whole body dose of greater than 1.5 Gy, as it may help identify persons who should be triaged to a medical facility having expertise in managing cytopenias. This test is not widely available and may be dependent upon the availability of a laboratory capable of performing this analysis [34].

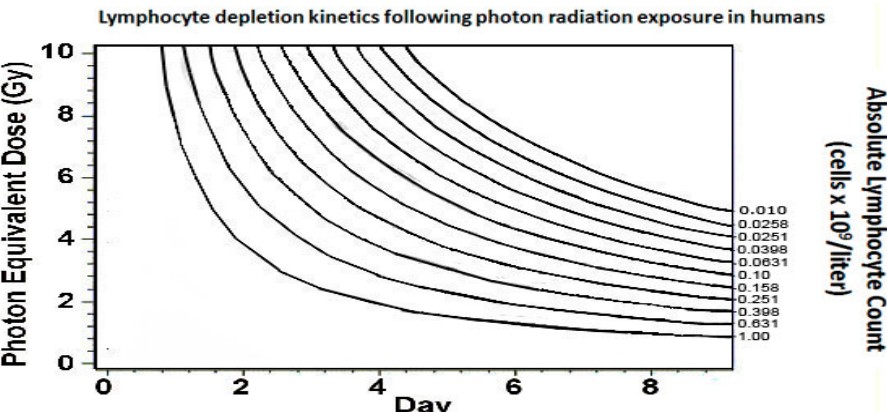

**Figure 4.** Health and Human Services. Radiation Event Medical Management. Source: www.remm.hhs.gov, accessed on 10 August 2023 [33].

Leukopenia, specifically neutropenia, will result in immunosuppression and impaired wound healing. Neutropenic patients will be at significant risk of both community acquired and nosocomial infections. Patients who have indwelling central vascular catheters are at very high risk for sepsis and should be aggressively worked up and treated in the event of fever, rigors, or hemodynamic instability. The patient should be kept in an isolation room (positive pressure if able), avoid interaction with the public or individuals known to have communicable illness, and wear a surgical mask when required to go into a public area. Patients who are neutropenic, defined by an absolute neutrophil count (ANC) of 500/uL or less, should be started on a myeloid cytokine such as filgrastim or pegfilgrastim. These cytokines should continue until the ANC is greater than 1000/uL for at least 3 days [35].

Anemia may require transfusions of packed red blood cells or whole blood. Blood products given to these patients should be irradiated and leuko-reduced to prevent transfusion-associated graft-versus-host disease, a rare but often fatal complication of transfusion in an immunocompromised patient. This syndrome is characterized by fever, rash, elevated hepatic enzymes, pancytopenia, and diarrhea [36]. Thrombocytopenia and subsequent bleeding risk poses a final problem for clinicians. Typical platelet transfusion parameters are less than $10 \times 10^3$/uL for an otherwise healthy patient, and less than $50 \times 10^3$/uL for a surgical patient. Patients requiring surgery while thrombocytopenic may require higher parameters depending upon the bleeding risk of a given procedure. Infusing platelets perioperatively can be preventive and is a strategy used by many hematopoietic stem-cell transplantation centers. In a resource-constrained environment, these parameters may need to be adjusted or disregarded completely.

Hematopoietic syndrome may be prolonged (weeks to months). The longer a patient requires myeloid cytokines and transfusions, the worse the prognosis. It is recommended that patients receive consultation by a hematologist or hematopoietic transplantation center if able, as transplantation may be the only chance for survival. This process is extensive and time consuming, requiring advanced testing for human lymphocyte antigens to determine donor compatibility. It is recommended that this process be initiated as early in the patient's care process as possible.

Neurovascular symptoms occur only in whole body doses in excess of 10 Gy. Neurovascular syndrome occurs in very high absorbed doses in the 10–20 Gy range. Symptoms present hours to days after the event and indicate a very poor prognosis, generally being fatal within days. Symptoms include severe nausea and vomiting, headache, unexplained cognitive and neurologic deficits, ataxia, and hypotension resulting from cerebral edema [32].

## 10. Countermeasures (Antidotes)

Decorporation therapy is the process of administrating an agent to decrease the risk of biological effects of a radionuclide which has been inhaled or ingested. Treatment of internal

contamination is based on the radionuclides involved and should occur in consultation with a professional who is knowledgeable about treating radiological injuries such as a hospital radiation safety officer, nuclear medicine physician, radiation oncologist, and/or a toxicologist [37]. The use of most countermeasures is off-label and carries an unknown risk-to-benefit ratio. For low levels of contamination, it is likely that risk outweighs the benefit [38]. Additionally, most authorities do not recommend treatment of internal contamination when the body burden is less than one annual limit of uptake [37]. Common countermeasures and their respective isotopes and mechanism of action are listed in Table 2.

**Table 2.** Select isotopes, their respective decorporating agents and mechanisms of action for internal radioactive contamination.

| Countermeasure | Isotope | Mechanism |
|:---:|:---:|:---:|
| Potassium Iodide | I-131 | Blocking Agent |
| EDTA | Co-60 | Chelating Agent |
| Prussian Blue | Cs-137 | Ion Exchange |
| Sodium Bicarbonate | U-235 | Facilitates renal excretion |
| Deferoxamine | Pu-239 | Chelating Agent |

Following a nuclear blast, there are several isotopes that are released into the environment which become concerns for internal contamination. Given that decorporating agents are specific to a single isotope, there is a public health need for rapidly identifying and quantifying the incorporated isotope and assessment of the associated committed dose so that medical countermeasures can be given as soon as possible [38]. The National Council on Radiation Protection and Measurements (NCRP) published clinical decision guidance (NCRP Report No. 166) which addresses this complex issue. In the setting of a RDD, the decorporation question is slightly less problematic due to the high probability of a single isotope. Nonetheless, it is difficult to assess the fraction of potentially contaminated victims that actually need treatment [39]. Using the "urgent" approach of the liberal dispensing of antidotes may save more lives in large scale events in which capacity for screening is low; however, this approach requires significantly greater stockpiles of antidotes. The "precautionary" approach of screening and treating only those that have sufficient internal contamination is the most efficacious way of reducing antidote requirements but requires the ability to screen large populations [39].

A nuclear reactor incident is a scenario in which a population may benefit from empiric antidote treatment. A historical example is the Chernobyl reactor criticality. The isotope I-131 would be the most notable contaminant in the environment from a reactor explosion. It is noted that following the Chernobyl incident, there were 7000 excess thyroid cancers in children and adolescents living in Ukraine, Belarus, and Russia proximal to the accident site. This is a 100 fold increase in incidence [40]. Stockpiles of readily available potassium iodide distributed to the exposed population likely would have significantly blunted this increase.

## 11. Conclusions

A large-scale radiation incident poses many clinical challenges. Hospitals can expect mass casualties that will stress radiation, blast, burn, and trauma services. Burn and trauma victim treatment will be complicated by exposure to ionizing radiation, both from the initial blast and subsequent fallout. Patients coming from a blast zone will likely need to be decontaminated and cared for by teams that are familiar with radiation meters and the decontamination process. Traumatic injuries are the number one priority in an irradiated patient and proficiency with providing lifesaving interventions in PPE will save numerous lives in a large-scale event. The presence of a combined injury, in particular, burn injuries of greater than 20% TBSA and an absorbed dose of radiation greater than 2 Gy, will significantly worsen the triage category. Burn surgical intervention should be

accomplished as soon as possible to avoid the complicating conditions posed by ARS; burn centers should maintain a close working relationship with hematology, oncology, and bone-marrow transplantation services to manage these patients. During this examination of the extreme public health, medical, and surgical complications that are associated with a nuclear event, it is apparent that significant efforts need to be made by the medical community to influence the prevention of these catastrophic events.

**Funding:** This research received no external funding.

**Institutional Review Board Statement:** Not applicable.

**Informed Consent Statement:** Not applicable.

**Data Availability Statement:** Not applicable.

**Conflicts of Interest:** The author declares no conflict of interest.

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
