# Peer review of "Management of Casualties from Radiation Events"

_2673-1991, doi:10.3390/ebj4040039_

Round 1
Reviewer 1 Report
Comments and Suggestions for Authors
In this manuscript, the authors discuss the types of burn injuries in a nuclear disaster, caring for the contaminated patient, and managing the combined injury of burn trauma with acute radiation syndrome. The reader will also be left with an understanding about how to prioritize lifesaving interventions by safely operating in a contaminated environment if necessary. However, this manuscript needs to be revised. A large number of discussions and descriptions lack references. And there is no comprehensive description of the new progress in the field. In addition, the organization of this manuscript is poor.
1.Title.
1) Line1.
The type of this manuscript should be a Review, not an Article.
2) Line2.
The title of this manuscript is Nuclear Preparedness, a phrase composed of two words, but it is overly broad and non-specific. The title will make people feel that this manuscript is a detailed guideline like National stockpiles for radiological and nuclear emergencies: policy advice published by the World Health Organization (WHO), but in fact this manuscript mainly focuses on wounds and burn injury induced by nuclear/ionizing radiation. The author should better revise the title.
2.Abstract.
1) The abstract of this manuscript is too short (100 words), and it is suggested to expand. Specifically, the background has been well introduced, but the abstract should also summarize the current gaps or problems in this field (radiation burn injury management). The summary should be critical and constructive and provides recommendations for future research.
2)Line6.
It would be more comprehensible for non-native readers to replace “nuclear reactor compromise” with “nuclear reactors leakage”.
3.Introduction.
1) Line19-23.
References are required.
2) Line23-25.
References are required.
4.Radiation Physics.
1) More references are required in this section.
2) Line33-36. ”There are... management.”
The description is not scientifically rigorous enough. First, the scope of "radiation incident" in this manuscript should be clarified or determined, because the types of ionizing radiation which impact health could be photons(X-rays or γ-rays), protons, electrons(β particles), α particles, neutrons or heavy ions depending on different radiation events. Second, neutrons are considered to be the main component of initial nuclear radiation, especially in neutron bombs explosion.
3) Line34.
Change “x-rays” to “X-rays”
4) Line46-56.
This paragraph mainly talks about β particles, but the example given is Cs-137, which is known to emit not only β particles but also γ-rays. It is suggested to replace with a more suitable case.
5) Line69-71.
When we talk about the acute lethal dose of radiation, the type of the radiation and the tested subjects/species should be clearly defined.
5.Nuclear Weapons.
1) More references are required in this section.
2) Line91-92
References are required. More researches about nuclear explosion dose reconstruction are suggested to cite.
6.Types of Burn Injury in a Nuclear Disaster.
1) Line111-112.
Supportive references should be cited.
2) This section is aimed to discuss the burn injury in a nuclear disaster, but the author mainly focuses on cutaneous radiation syndrome(CRS) and also mention about combined injury, both are not classified as single burns. It is known that the burn injury is usually induced by the light/thermal radiation, but CRS is induced by initial nuclear radiation, radioactive contamination and so on. The author should better discriminate/distinguish these concepts for readers and a simple table is suggested to be used to summarize the types of burn injury in a nuclear disaster.
3) Recent years, some new treatments against CRS have been developed, e.g stem cell therapy, if related discussions could be conducted, the quality of this manuscript will be improved. Besides, the author should better refer to clinical practice guidelines and some drugs already on the market.
7.Radioactive Dispersal Device (RDD).
1) The author is suggested to cite some cases or studies to describe and discuss the possible dose caused by RDD or dirty bombs.
e.g https://doi.org/10.1186/s40779-021-00349-w
8.Acute Radiation Syndrome.
1) How dose ARS influence/affect CRS?
9.Countermeasures (antidotes).
1) The content is too general.
Author Response
Thank you for your comments!
- Changed to "review", Title changed to "Management of Casualties from Radiation Events"
- Abstract expanded as recommended, reactor compromise changed to reactor incident here and throughout the review
- References added
- provided explanation of additional radiation forms, neutrons and X-rays, with explanation as to their role
- beta case changed to Chernobyl beta burn case
- Lethal dose subject / species added
- Nuclear weapons references added
- Burns - expanded material on burns unique to nuclear weapon. Clarified CRS and CRI, added novel treatments,
- RDD dose reconstruction from a rad perspective added
- CRS / ARS relationship discussed in 168-170
- Countermeasure section was expanded, I didn't want to get to specific or detailed because it's a complex sometimes controversial problem and not the focus of this paper.
Thank you again.
Reviewer 2 Report
Comments and Suggestions for Authors
is there a place to write in more detail about preparedness if there is a way? in a national level and disaster planning are there any relevant articles? it is a risk evaluation that political issues might interfere....
Author Response
Thank you for your comments.
Based on your input I added a comment about preparedness (as a need) in the abstract. Preparedness is a whole section in itself and given the word count I am limited to, it would be difficult to add that section without taking out key medical information.
Reviewer 3 Report
Comments and Suggestions for Authors
I think this is an extremely well written and highly relevant paper that very clearly introduces the reader to the fundamentals of nuclear incident management from a clinical perspective. It covers very well the background science and pathophysiology in straightforward and easy to understand language. I only have a couple of very minor comments:
1. P5 line 212/213 'The general rule is that if a patient has a combined injury the triage category would be elevated 1-2 levels' - is there any evidence for this statement and if not from where has this 'general rule ' come
2. P 9 Line 346-351...there will probably be no major role for the use of decorporating agents...This seems to go slightly against the evidence in that one of the very very few nuclear events that have occurred (Chernobyl) it seems to be clear that providing potassium iodide may have helped prevent future malignancy? As a cheap, generally available substance with minimal side effects would it not be appropriate to widely prescribe this?
On a final point I am slightly surprised the paper 'Justification for a Nuclear Global Health Force: Multidisciplinary analysis of risk , survivability and preparedness, with emphasis on the triage management of thermal burns' - Burkle et al. Conflict and Health (2017) 11:13 was not cited at all in the references
Author Response
Thank you for your comments.
I reworded the "general rule" phrase and added a reference for elevation of triage categories
I clarified the decoporation section and better described the problematic nature of who actually needs antidote especially after a nuclear blast, I also added your recommended example of Chernobyl and providing KI would have potentially prevented hundreds of cases of thyroid cancer.
Thank you for the reference (Burkle et al) . I did overlook this and have since added it
Thank you again